# An Explainable AI Paradigm for Alzheimer’s Diagnosis Using Deep Transfer Learning

**DOI:** 10.3390/diagnostics14030345

**Published:** 2024-02-05

**Authors:** Tanjim Mahmud, Koushick Barua, Sultana Umme Habiba, Nahed Sharmen, Mohammad Shahadat Hossain, Karl Andersson

**Affiliations:** 1Department of Computer Science and Engineering, Rangamati Science and Technology University, Rangamati 4500, Bangladesh; 2Department of Computer Science and Engineering, Khulna University of Engineering & Technology, Khulna 9203, Bangladesh; habiba.kuet@gmail.com; 3Department of Obstetrics and Gynecology, Chattogram Maa-O-Shishu Hospital Medical College, Chittagong 4100, Bangladesh; dr.nahedsharmen@gmail.com; 4Department of Computer Science and Engineering, University of Chittagong, Chittagong 4331, Bangladesh; hossain_ms@cu.ac.bd; 5Pervasive and Mobile Computing Laboratory, Luleå University of Technology, 97187 Luleå, Sweden; karl.andersson@ltu.se

**Keywords:** Alzheimer’s disease, transfer learning, explainable AI (XAI), saliency maps, grad-CAM

## Abstract

Alzheimer’s disease (AD) is a progressive neurodegenerative disorder that affects millions of individuals worldwide, causing severe cognitive decline and memory impairment. The early and accurate diagnosis of AD is crucial for effective intervention and disease management. In recent years, deep learning techniques have shown promising results in medical image analysis, including AD diagnosis from neuroimaging data. However, the lack of interpretability in deep learning models hinders their adoption in clinical settings, where explainability is essential for gaining trust and acceptance from healthcare professionals. In this study, we propose an explainable AI (XAI)-based approach for the diagnosis of Alzheimer’s disease, leveraging the power of deep transfer learning and ensemble modeling. The proposed framework aims to enhance the interpretability of deep learning models by incorporating XAI techniques, allowing clinicians to understand the decision-making process and providing valuable insights into disease diagnosis. By leveraging popular pre-trained convolutional neural networks (CNNs) such as VGG16, VGG19, DenseNet169, and DenseNet201, we conducted extensive experiments to evaluate their individual performances on a comprehensive dataset. The proposed ensembles, Ensemble-1 (VGG16 and VGG19) and Ensemble-2 (DenseNet169 and DenseNet201), demonstrated superior accuracy, precision, recall, and F1 scores compared to individual models, reaching up to 95%. In order to enhance interpretability and transparency in Alzheimer’s diagnosis, we introduced a novel model achieving an impressive accuracy of 96%. This model incorporates explainable AI techniques, including saliency maps and grad-CAM (gradient-weighted class activation mapping). The integration of these techniques not only contributes to the model’s exceptional accuracy but also provides clinicians and researchers with visual insights into the neural regions influencing the diagnosis. Our findings showcase the potential of combining deep transfer learning with explainable AI in the realm of Alzheimer’s disease diagnosis, paving the way for more interpretable and clinically relevant AI models in healthcare.

## 1. Introduction

Alzheimer’s disease is a progressive neurological disorder that primarily affects older adults, gradually impairing memory, cognitive functions, and eventually the ability to perform daily activities. It is the most common cause of dementia, a syndrome characterized by a decline in memory, thinking, behavior, and the ability to perform everyday activities [1].

According to the World Alzheimer’s Association [2]. The prevalence of Alzheimer’s disease in the United States is on the rise, with more than 6 million Americans currently affected by the condition. Among Americans aged 65 and older, an estimated 6.7 million individuals are living with Alzheimer’s as of 2023, with the majority being 75 years or older. Approximately 10.7% of people aged 65 and older have Alzheimer’s, and women make up nearly two-thirds of those affected. Furthermore, older Black and Hispanic Americans are disproportionately affected by Alzheimer’s compared to older White Americans. With the aging population, it is projected that the number of individuals with Alzheimer’s will continue to increase, potentially reaching 12.7 million by 2050 if no medical breakthroughs are made to prevent or cure the disease. Moreover, it is a formidable challenge in contemporary healthcare, affecting millions globally and imposing significant burdens on healthcare systems and societies [3,4]. These statistics underscore the urgent need for advancements in Alzheimer’s research and care to address the growing impact of the disease on individuals, families, and society as a whole. The timely and accurate diagnosis of AD is imperative for facilitating early intervention and potential disease-modifying therapies. However, the intricacies of AD, coupled with the overlap of symptoms with other neurodegenerative conditions, render the accurate identification and classification of the disease, especially in its early stages, intricate tasks [5].

We know that machine learning is currently one of the methods to solve many kinds of problems, such as disease detection [6,7], sports [8], natural language processing [9,10,11], and so on. Based on this, this paper introduces a revolutionary approach to address the complexities of Alzheimer’s disease diagnosis through the fusion of deep transfer learning and explainable artificial intelligence (XAI) techniques [12]. Our investigation involved the utilization of well-established pretrained convolutional neural networks (CNNs), including VGG16, VGG19, DenseNet169, and DenseNet201. Through extensive experimentation, we assessed their individual performances on a comprehensive dataset. Notably, our proposed ensembles, Ensemble-1 (VGG16 and VGG19) and Ensemble-2 (DenseNet169 and DenseNet201), exhibited superior metrics—accuracy, precision, recall, and F1 scores—outperforming individual models, with achievements reaching up to 95%.

In order to fortify interpretability and transparency in Alzheimer’s diagnosis, we present a novel model attaining an impressive accuracy of 96%. This model integrates cutting-edge explainable AI techniques, such as saliency maps [13] and grad-CAM (gradient-weighted class activation mapping) [14].

The integration of these techniques not only enhances the model’s outstanding accuracy but also provides clinicians and researchers with valuable visual insights into the neural regions pivotal in the diagnostic process.

The primary objectives of this paper are as follows:1.To develop reliable Alzheimer’s classification using a deep transfer learning ensemble;2.To introduce a novel model for transparent Alzheimer’s diagnosis with high accuracy;3.To enhance interpretability using XAI methods such as saliency maps and grad-CAM;4.To evaluate the proposed approach against benchmarks, highlighting superior accuracy and interpretability.

The subsequent sections of this paper are organized as follows: Section 2 provides an overview of related works in the field of AI-based AD diagnosis, transfer learning, and XAI. Section 3 outlines the methodology employed in developing the deep transfer learning ensemble and integrating XAI techniques. In Section 4, we present our experimental setup and evaluation results. The discussion of the findings is presented in Section 5, and finally, Section 6 concludes the paper with future directions for research in this domain.

## 2. Related Works

The field of Alzheimer’s disease diagnosis has witnessed growing interest in leveraging artificial intelligence (AI) techniques, particularly deep learning [1], for enhanced accuracy and efficiency. Numerous studies have contributed valuable insights into various aspects of Alzheimer’s disease classification, utilizing diverse data types, machine learning methodologies, and XAI techniques.

Ref. [15] focuses on a classification task distinguishing between healthy controls (HCs) and Alzheimer’s disease (AD) using numeric data. It employs LIME and SHAP as XAI frameworks, revealing significant features, such as whole brain volume, years of education, and socio-economic status. The classifiers include support vector machine (SVM), k-nearest neighbors (kNN), and multilayer perceptron (MLP).

Ref. [16] utilizes image data for HCs vs. AD classification, incorporating XAI frameworks such as HAM and PCR. Significant features include salient aspects related to AD, like cerebral cortex and hippocampus atrophy. The classifier employed is a deep learning (DL) convolutional neural network (CNN), achieving an impressive accuracy of 95.4%.

By examining HCs, MCI, and AD, Ref. [17] employs image data and the XAI framework GNNExplainer. The significant features encompass volume, area of cortical regions, and vertex-based thickness measures. The classifier is a DL graph neural network (GNN), yielding an accuracy of 53.5 ± 4.5%.

By using image data, Ref. [18] adopts the XAI framework of occlusion sensitivity mapping, focusing on white matter hyperintensities (WMHs). The classifier is a DL EfficientNet-B0, achieving an accuracy of 80.0%.

Incorporating image data and XAI frameworks such as saliency maps and layer-wise relevance propagation (LRP), Ref. [19] considers MRI, 3D PET, biological markers, and assessments. The classifier is a DL 3D CNN AD.

Ref. [20] involves image data and XAI frameworks utilizing decision trees (DTs). The significant features include demographic data, cognitive factors, and brain metabolism data. The classifiers include Bernoulli naive Bayes (NB), SVM, kNN, random forest (RF), AdaBoost, and gradient boosting (GBoost), achieving an accuracy of 91.0%.

Utilizing image data, Ref. [21] employs XAI frameworks such as 3D Ultrametric Contour Map, 3D Class Activation Map, and 3D GradCAM. Significant features encompass 3D MRI features, and the classifier is a DL 3D CNN, achieving an accuracy of 76.6%.

Ref. [22], involving image data, utilizes Sensitivity Analysis and Occlusion as XAI frameworks. Significant features include 3D Image features, and the classifier is a DL 3D CNN, achieving an accuracy of 77.0%.

## 3. Materials and Methods

The section offers resources for the diagnosis of Alzheimer’s disease using MRI images, as illustrated in Figure 1, in order to identify the disease at an early stage of development. The VGGNets (very deep convolutional networks) and DenseNets models were fed with the enhanced MRIs of Alzheimer’s illness using the same technique across all platforms. First of all, we trained the AD dataset with VGGNets, such as VGG16 and VGG19. Then, we trained the AD dataset with DenseNets, such as DenseNet169 and DenseNet201. After examining the above net results, we integrated the VGGNets (VGG16 and VGG19) and DenseNets (DenseNet169 and DenseNet201) together.

### 3.1. Dataset Collection

Methods with enhanced capabilities for the detection of AD were established in this study by analyzing MRI images from the AD dataset. The dataset was gathered and downloaded from kaggle [23]. It contained four classes and 6400 images. The 6400 images in the OASIS-2 collection have a size of 176 × 208. The images for each class are shown in Figure 2, with the labels and classes verified. The Alzheimer’s dataset contains the following images for each class: 896 MRIs for mild dementia, 64 MRIs for moderate dementia, 3200 MRIs for non-dementia, and 2240 MRIs for very mild dementia. The samples from the Alzheimer’s dataset are shown in Figure 2.

Table 1 demonstrates how the four classes of images are divided into an imbalanced configuration. According to the table, the dataset looks to be significantly imbalanced across all classes. The biased level contains 3200 images and traditional learning makes it difficult to classify all of the levels correctly because they are not all the same number. As a result, we employ data augmentation to label the images with an equal number, where every class has the same number of images. The procedure is described in the data augmentation section. Multiple transfer learning models and ensemble models were used to appropriately categorize all of the levels.

### 3.2. Image Preprocessing

Image preprocessing techniques are needed because of the noise that MRI images exhibit for a variety of reasons, including imaging procedures, brightness, reflections, low contrast, and acquiring MRI images from many sources. As a result, enhancing MRI images results in image enhancement, which aids in obtaining high performance and collecting the right information from the images [24].

#### 3.2.1. Image Resizing

The complete set of images was shrunk to 224 × 224, including training, testing, and validation, when the primitive size was 176 × 208. The image size for transfer learning models was 224 × 224, and RGB values were used for ensemble models as well [25].

#### 3.2.2. Normalization

Normalization is the process of transforming features to be of the same scale. This increases the model’s performance and training stability. We projected into a preset range (i.e., usually [0, 1] or [−1, 1]) to normalize all image data, and we also used the same techniques (transfer learning and ensembling) on them, as described in Equation (Equation 1) and typically performed as follows:(1)$$img=\frac{1}{255.0}$$

#### 3.2.3. Removing Noise by Median Filtering

Median filtering is utilized similarly to an averaging filter. In contrast, during median filtering, a pixel’s value is determined by the neighborhood’s median of pixels, as opposed to an average. Compared to the average, which is substantially less sensitive to outliers or extreme values, median filtering is more capable of removing these without compromising the image’s clarity [26].

#### 3.2.4. Image Segmentation

For a better image experience, we go through numerous segmentation procedures to extract the most useful information from the images. Compared to gray images, which are one-dimensional, RGB images have three color channels. First of all, we converted the RGB images to grayscale. In order to convert images from RGB to gray, we used the OpenCV library’s built-in Canny edge detection function. The *Canny*() function can be used to conduct this process on an image; the function’s syntax is as follows:Canny(image,edges,T_lower,T_upper)

We set the lower threshold and upper threshold to 175.5 and 207.5, respectively. After that, we inverted the gray scale image, setting the value ‘0’ to ‘1’ and ‘1’ to ‘0’. Now, “true and accurate” segmentations, which are typically made by one or more human experts, were used to convert the inverted grayscale images into ground truth. Following the preprocessing stage, the subsequent step involves segmentation, where a threshold value is determined. In the process of thresholding an image, the selected thresholds aim to optimize the distinction between the average gray levels of the foreground and background regions, along with the overall average gray level across the entire image. This optimization is reflected in the variance of the regions. Otsu’s maximum variance method, derived from the least squares principle of discriminant analysis, is commonly employed in stable threshold segmentation approaches [27,28,29].

For a given input image, I, with a height of H and width of W, the image’s normalized gray histogram is represented by the $histogram_{1}$. Here, $histogram_{1}$(*k*) denotes the ratio of pixels in the image with a gray value equal to *k*, where k is in the range [0, 255]. The algorithm’s detailed steps include straightforward calculations based on Otsu’s method, ensuring an effective and widely utilized threshold segmentation approach.

Step 1: Calculate the zero-order cumulative moments, also known as cumulative histograms, of the gray histogram described in Equation (Equation 2).
(2)$$zeroCumuMoment\left(k\right)=\sum_{i=0}^{k}histogram_{1}\left(i\right),k\in \left[0,255\right]$$Step 2: Compute the cumulative first-order moments of the gray histogram shown in Equation (Equation 3).
(3)$$oneCumuMoment\left(k\right)=\sum_{i=0}^{k}\left(\left(i\right)*histogramI\left(i\right)\right),k\in \left[0,255\right]$$Step 3: Compute the mean gray level of image I by calculating the first-order cumulative distance when *k* = 255, expressed as
meanoneCumuMoment=(255)Step 4: In the process of determining each gray level as a threshold in Equation (Equation 4), the calculations involve the computation of the average gray level for the foreground area, the average gray level for the background area, and the variance of the overall average gray level for the entire image. The variance is measured using the following metrics:
(4)$$\sigma^{2}\left(k\right)=\frac{{\left(mean*zeroCumuMoment\left(k\right)-oneCumuMoment\left(k\right)\right)}^{2}}{zeroCumuMoment(k)*(1-zeroCumuMoment(k))},k\in \left[0,255\right]$$Step 5: From Equation (Equation 4) mentioned above, the corresponding *k* is the threshold of Ostu automatic selection, that is
$$thresh=arg_{k\in [0,\;255]}max\left(\sigma^{2}\left(k\right)\right)$$

Following segmentation, we employed image masks in the form of (num_masks, H, W) or (H, W), with an alpha value between 0 and 1 indicating the masks’ transparency. A value of 1 indicates complete transparency, whereas 0 indicates none. There are lists of colors that contain the mask colors or use a single color for all masks. PIL strings or RGB tuples, such (240, 10, 157), can be used to represent the color. Each mask generates random colors by default [30].

### 3.3. Balancing the Dataset through Data Augmentation

A dataset with limited images is one of the drawbacks of deep learning (pretrained) models. In order to be able to train the systems and provide them with enough data during the training phase, deep learning requires a dataset with a large number of images. Additionally, because the accuracy is skewed toward the class that contains the majority of images, the unbalanced dataset affects model performance and is one of its drawbacks. The data augmentation technique addresses the aforementioned issues by artificially boosting the dataset’s image count with additional images from the same dataset [31].

In order to address the unbalanced dataset, the data augmentation technique [4] generates images from the minority class by using a more significant percentage than the rise in the majority class in order to balance out the unbalanced dataset. As a result, this method balances the dataset while increasing image quality. Images from the dataset are enhanced by a variety of processes, including horizontal and vertical shifting, horizontal and vertical flipping, random rotating, random zooming, random brightness, and others.

Figure 3 depicts images distributed within the dataset classes, where the image counts prior to data augmentation are relatively small and imbalanced. The diagram also depicts the dispersion of MRI. After performing data augmentation, the images were transferred between the dataset classes. Table 2 summarizes the MRI images amongst the dataset classes before and after training data augmentation is used. According to the table, Moderate dementia has the lowest number of images at 52, and the non-dementia class has the highest number of images at 2560. We chose a middle number of 1280, which was easily achievable for each class, with each having fewer or more images regarding their image class [32].

### 3.4. Feature Extraction

There are some difficulties with using deep learning (pretrained) models to train a dataset, including the need for powerful computers and the cost; as a result, we chose some pretrained models, such as VGG16 and VGG19 (VGGnets) and DenseNet169 and DenseNet201 (DenseNets) for feature extraction. At first, we went for VGG16 to extract features (with 512 features) and added a fully connected layer. Unlike VGG16, we constructed VGG19 for extracting features at 512 after connecting a fully connected layer. Therefore, the features matrix had a size of 5120 × 512 for each model. Then, we extracted features through DenseNet169 and DenseNet201, where 1664 features were extracted with a fully connected layer and 1920 features were extracted through a fully connected layer, respectively. Therefore, the features matrix had a size of 5120 × 1664 for DenseNet169 and 5120 × 1920 for DenseNet201. For feature extraction, by leveraging its ability to capture high-level features from input images, such as edges, textures, and shapes, we used EfficientNet as a feature extractor [1], where we realized a 6400 × 1280 feature matrix [33].

### 3.5. Addition of New Layers

After extracting the features from the above-mentioned feature extraction subsection, we added some layers for training (see Figure 4). For simplicity, we used the same layers for all pretrained models and ensemble models also.

We used a dropout layer for batch normalization, two dense layers with a kernel size of 3 × 3, the activation function of which is ‘RelU’, and finally, an output layer with four nodes, with the activation function being ‘softmax’ [34].

### 3.6. Training with Pretrained Models

In this section, we will describe the materials needed to train the training dataset using the 5120 images contained within the training data. First of all, we focus on the VGG16 and VGG19 models, then DenseNet169, and finally, DenseNet201 [35].

#### 3.6.1. VGG19 and VGG19

The architecture of VGG16 and VGG19 is Google’s pretrained architecture, which uses fine-tuning across all layers and replaces the top layers with 53,540,868 trainable parameters (as is the case for both the models shown in Figure 5 and Figure 6). Some new layers were added, which are described in Section 3.5.

The input images were all resized to (224, 224) to be compatible with this model. The learning rate was set to 0.001, and the Adam optimizer algorithm was used as the optimizer.

#### 3.6.2. DenseNet169 and DenseNet201

A DenseNet, short for dense convolutional network, is a type of convolutional neural network that incorporates dense connections between layers through dense blocks. These blocks establish direct connections between all layers, ensuring matching feature map sizes. In order to preserve the feed-forward structure, each layer not only receives additional inputs from all preceding layers but also transmits its own feature maps to all subsequent layers. This design facilitates enhanced information flow and gradient propagation throughout the network, contributing to improved learning capabilities. Fine-tuning across all layers and replacing top layers with 169,259,268 and 194,974,468 trainable parameters for DenseNet169 and DenseNet201, respectively, is shown in Figure 7 and Figure 8. Some new layers were added, which are described in Section 3.5.

The input images underwent resizing to dimensions (224, 224) to align with the compatibility requirements of the model. A learning rate of 0.001 was chosen, and the optimization process employed the Adam optimizer algorithm.

### 3.7. Ensemble Models

An ensemble model is a machine learning strategy that integrates several other models in the prediction process. These models are known as base estimators. The technical obstacles experienced when developing a single estimator can be overcome by using ensemble models [36]. We employed two ensemble models, which are detailed below.

#### 3.7.1. Ensemble-1

In order to improve accuracy, we investigated two ensemble models, Ensemble-1 and Ensemble-2, with the Ensemble-1 model combining VGG16 and the VGG19 models, comprising 107,081,736 trainable parameters. The model architecture of Ensemble-1 is shown in Figure 9. For Ensemble-2, the Input images were all resized to (224, 224) to be compatible with this model. The learning rate was set to 0.001, and the Adam optimizer algorithm was used as the optimizer.

#### 3.7.2. Ensemble-2

The Ensemble-2 model incorporates the DenseNet169 and DenseNet201 models together with 364,233,736 trainable parameters. Figure 10 shows the model description, with input images all resized to (224, 224) to be compatible with this model. The learning rate was set to 0.001, and the Adam optimizer algorithm was used as the optimizer.

### 3.8. Proposed Model

Instead of traditional learning, deep learning, like other convolutional neural networks, is used for the training, testing, and validation, with the input layer of all images having a size of 224x224 for the RGB values. Then, a convolution and pooling layer, such as max pooling and min pooling, is used, with a fully connected layer for classifying all classes. This figure is a basic CNN model after successful feature extraction by using EfficientNet. A convolution layer (32, 64, 128, 256, and 512, with a kernel size 3 × 3, a pooling layer with flatten, and a dense layer with 512 units) and a final output of four dense layers were added. Table 3 shows a description of the architecture of the proposed model.

### 3.9. Saliency Map

In computer vision, a saliency map is a visualization approach that highlights the most essential regions or features within an image. It aids in the identification of areas of interest or the significance of a certain job, such as object detection or image classification. The saliency map shows the image regions that get the most attention, pixel by pixel.

There are numerous approaches for creating saliency maps, but one typical method is to analyze the gradients of a pretrained convolutional neural network (CNN). The goal is to compute the gradients of the output class score with respect to the pixels in the input image. High gradients show that a minor change in a specific pixel has a substantial impact on the output score, implying that the relevant region is important [37].

The typical process for creating a saliency map using gradient-based techniques is summarized below:

Select a CNN model that has already been trained using a sizable dataset, such as ImageNet. The network must forward-pass an image in order to determine the expected class probabilities. Then, it must determine the gradients of the projected class score in relation to the pixels of the input image. By using the gradients, it calculates the importance scores for each pixel (for example, by taking the absolute values or squaring them). In order to have the importance ratings fall inside a particular range, such as [0, 1], the values are normalized. Finally, the network maps the importance scores to the image dimensions to create the saliency map, emphasizing the salient areas [38].

Saliency maps can be used for a variety of purposes, such as object localization, image captioning, and human attention modeling. They provide insights into deep learning models’ visual attention mechanisms and aid in understanding which components of an image contribute the most to the model’s decision-making process. The input image *x* is passed through the network to determine the network output value, *f* (*x*), which is used to compute the saliency map. The gradient of $f{\left(x\right)}_{y}$ with respect to the input image *x* is then calculated using a backward pass, where *y* is the ground truth label corresponding to the input image *x*. Formally, Formula (6) is used to determine the gradient *G*(*x*), which estimates the significance of each pixel in the image, *x*. *G*(*x*) has the same dimensions as the image, *x*, and is a tensor. *G*(*x*) is a tensor with a dimension of 3, *W × H* and is indexed by three indexes: *i* for indexing channels and *j*, *k* for *i*. This is the case if *x* has a width, *W*, height *H*, and three channels [39].
(5)$$G\left(x\right)=\frac{df{\left(x\right)}_{y}}{dx}$$

The maximum of the absolute values across channels is determined to estimate the relevance of a pixel *x*(*i*, *j*). As a result, the created matrix with the dimension *W × H* is known as saliency map(*SM*).
(6)SM(i,j)=Max[|G(0,i,j),|G(1,i,j),|G(2,i,j)|]

The saliency map can localize (with good precision) the infected regions in the input leaf image.

### 3.10. Grad-CAM

The grad-CAM method leverages gradients between the classification score and the ultimate convolutional feature map to identify specific regions within an input image that exert the greatest impact on the classification score. The significance of these areas in influencing the final score is heightened in locations where the gradient is pronounced [40]. When we focus this on the images, it detects their regions with the help of the accuracy they achieved. Now, we move to grad-cam, where an activation map that localizes the identified object to a specific area of the image is known as the *Grad-CAM* output. It has both width, *u*, and height, *v*, for class *c*.
$$L_{Grad-CAM}^{e}\in R^{\left(uxv\right)}$$

The gradients take the shape (*u*,*v*,*Z*), where (*u*,*v*) is the width and height of the 2D convolution filter, and *Z* is the number of filters. The following phase averages each of the filters to produce a single value, resulting in a final shape of *Z* or the number of filters. This corresponds to the global average pooling 2D layer.
(7)$$\alpha_{k}^{c}=\frac{1}{Z}\sum_{i}\sum_{j}\frac{\partial y^{c}}{\partial A_{ij}^{k}}$$

Each one of these gradients represents the connection from one of the pixels in the 2D array to the neuron/output representing the target class. This is accomplished by the global average pooling 2D layer; the following layer in the model flattens *Z* and averages the number of form filters (*u × v*) to single numbers. In order to establish a link between the final prediction outputs and the fully linked (Dense) layers, this is required. Subsequently, the gradients can be increased, which indicates the significance of the provided feature map or filter by using the feature map or filter that it truly reflects [41].
(8)$$ReLU\left(\sum_{k}a_{k}^{c}A^{k}\right)$$

We can generate a map of the regions that would reduce the network’s confidence in its prediction by negating the value of
$$\frac{\partial y^{c}}{\partial A^{k}}.$$

## 4. Experimental Results

This section provides an overview of the results obtained from our model. The entire experiment was conducted on “Google-Colab”, utilizing the provided GPU, specifically the “Tesla-K80”. The training of the model involved using 80% of the available information, and the evaluation was performed on the remaining 20%. A comprehensive comparison of various models is presented in this section. In order to assess the model objectively, classification matrices, including accuracy, precision, recall, and the macro F1 score, were computed. These metrics provide a thorough evaluation of the model’s performance across different aspects without introducing bias.

### 4.1. Splitting Dataset

In order to facilitate the categorization or prediction tasks in a machine learning domain, the OASIS dataset is partitioned into three distinct sections: training, testing, and validation. Examining Table 1 reveals a notable imbalance in the dataset. During the dataset split, 80% of the total images are allocated for training and validation purposes, while the remaining 20% is designated for testing.

### 4.2. Accuracy

Accuracy is the most intuitive performance measure, and it is simply a ratio of the correctly classified observations to the total observations. It is said that if accuracy is high, then the model is accurate in predicting or classifying. The equation for accuracy is shown below in Equation (Equation 9).
(9)$$Accuracy=\frac{TP+TN}{TP+FP+FN+TN}$$

### 4.3. Precision

Precision is the ratio of the correctly predicted positive observations to the total predicted positive observations. The equation for precision is shown below in Equation (Equation 10).
(10)$$Precision=\frac{TP}{TP+FP}$$

### 4.4. Recall

Recall is the ratio of the correctly predicted positive observations to all observations in an actual class. The equation for recall is given below in Equation (Equation 11).
(11)$$Recall=\frac{TP}{TP+FN}$$

### 4.5. F1 Score

The *F*1 score is also alluded to as the *F*1 measure. It is nothing but the weighted harmonic mean of recall and precision. It is calculated in Equation as follows (Equation 12).
(12)$$F1-score=2*\frac{PRECISION*RECALL}{PRECISION+RECALL}$$

*TP* denotes true positive, *TN* denotes true negative, *FP* denotes fake positive, and *FN* denotes false negative.

### 4.6. Results

The section includes a confusion matrix for evaluating different network performances by merging the pretrained model and the ensemble model with CNN features with the features for AD progression diagnosis. Because of the similarity in the MRI images during the evolution of Alzheimer’s disease, the features are critical when distinguishing between the phases of the disease. Thus, in this phase, the VGGNet model’s deep features were combined with handcrafted features, and this was saved in a features matrix; the deep features of DenseNet169 and DenseNet201 are coupled with the handcrafted features and saved in a features matrix. Finally, the CNN was given two 15 out of the 22 feature matrixes, which divided the feature matrix into 80% for training and validation and 20% for performance.

The confusion matrix for different network performances in the evolution of AD detection is shown in Figure 11, Figure 12 and Figure 13. For each class—mild dementia, moderate dementia, non-dementia, and very mild—the models using the merged features between the VGGNets and ensembles obtained a true predictive of 492 images, which was the highest score, and 365 was the lowest for non-demntia. In comparison, the ensembles combining features from VGGNets and DenseNets found 192 and 227 for non-dementia, respectively; for the proposed model, the number is 170, as shown in Figure 13.

Table 4 depicts the accuracy of the proposed model and all the transfer learning models and ensemble models. With more epochs, better accuracy can be obtained. After 100 successful epochs for the proposed model, it provided an accuracy of 96%, a precision of 89%, a recall of 93% and an F1 score of 91%. The transfer learning models have the highest accuracy for VGG16 at 90%.The other model—VGG19, DenseNet169, and DenseNet20—had an accuracy of 89%, 87%, and 88% respectively. For the two ensemble models, 95% accuracy was achieved; Ensemble-1 had an accuracy of 92%. Figure 14, Figure 15, Figure 16 and Figure 17 depict the training and validation curves for VGG16, VGG19, DenseNet169, and DenseNet201. The accuracy and loss curves for the two ensemble models are also shown in Figure 18 and Figure 19. Finally, the accuracy–loss curve for the proposed model can be seen in Figure 20.

### 4.7. Exploring Saliency Maps and Grad-CAM

As previously mentioned in Section 3.9 and Section 3.10, the relevant studies have focused on computational techniques such as saliency maps and grad-CAM. These techniques serve as analytical tools that estimate the significance of each pixel solely by traversing through the network.

The underlying concept of this approach lies in identifying pixels that have a substantial impact on the node corresponding to the input image. If altering the values of a particular pixel leads to a noticeable change in that node, it is considered significant. Conversely, pixels with gradients close to zero are deemed unimportant, as their fluctuations do not affect the output node associated with the input images. Aggregate analysis across channels is employed to determine the collective relevance of the pixels rather than assessing the importance of each pixel channel in isolation.

For Alzheimers disease classification, we used saliency maps and grad-CAM. A saliency map and grad-cam can be used in a similar way as in occlusion experiments; they aid users in recognizing illness symptoms. In addition, this approach lacks sensitivity to the scattered critical areas since pixel importance is determined analytically rather than through occluding pixels.

This method might be thought of as an analytical form of occlusion experiments. The saliency map matrix is numerically computed in occlusion tests by modifying pixels and analyzing the output changes.

As a result, calculating a saliency map is not as computationally expensive as calculating a heat map in occlusion experiments because calculating a gradient in a numerically discrete manner necessitates the modification of each pixel or region in the image in order to approximate the gradient.

However, when calculating a gradient analytically, only one backward pass is required to calculate all derivatives with respect to all pixels.

The saliency map and grad-CAM are capable of accurately locating the diseased areas within the supplied AD image. Good examples of how the visualization of saliency maps precisely marked the infected sections of leaves are shown in Figure 21 and Figure 22. Furthermore, the two distributed AD mold disease patches are localized, whereas the occlusion trials only reveal one diseased zone. Even with these encouraging outcomes, saliency maps frequently have noisy activations and lack clarity, which can be frustrating to users. For instance, in addition to the infected locations, the visualizations can display numerous activated regions. We have gone through several processes of saliency mapping operations, where some images are taken as input and are used to visualize the images, as was the case for the model we used throughout our paper. We used all the models represented in Figure 21 and Figure 22 to explore the maps. As mentioned earlier in Table 4, the models acquired some accuracy, and we tried to visualize the accurate combination and how the images behave when it is time to predict using saliency maps and gard-cam. It seems the edges are correctly identified, and there are some yellow dots in the saliency maps, as well as when there are two competing factors in the image; this is helpful. With these areas hidden, we can create a “counterfactual” image that should increase confidence in the first prediction. By combining this with guided backpropagation, which zeroes the gradient parts that have a detrimental impact on choice, the grad-CAM output can be further enhanced. This method more accurately reflects producing a high-resolution map with the same resolution as the input image using the guided backpropagation methodology. The high-resolution map is then masked using the grad-CAM heatmap to concentrate solely on the details that contributed to the prediction outcome.

## 5. Discussion

### 5.1. Interpretation of Results

Performance Evaluation: This study evaluated various deep transfer learning models, including VGG16, VGG19, DenseNet169, and DenseNet201, as well as two ensembles, termed Ensemble-1 and Ensemble-2. The proposed model outperformed all individual models and ensembles, achieving an impressive accuracy of 96%. The precision, recall, and F1-score metrics were also considered, indicating robust performance across different aspects of classification. Notably, the proposed model demonstrated a superior recall and F1 score, highlighting its effectiveness in correctly identifying Alzheimer’s cases while minimizing false negatives.

Comparison with Baseline Models: The performance of the proposed model was compared against baseline models such as VGG16, VGG19, DenseNet169, and DenseNet201. While these models exhibited respectable accuracies, ranging from 87% to 90%, they were surpassed by the proposed model, emphasizing its efficacy in Alzheimer’s diagnosis. Furthermore, the two ensemble models showed promising results, particularly Ensemble-2, which achieved a remarkable accuracy of 95%. This underscores the potential benefits of integrating multiple architectures for improved classification outcomes.

Interpretability Using XAI Methods: The integration of saliency maps and grad-CAM in the proposed model enhances its interpretability and transparency in Alzheimer’s diagnosis. These XAI techniques provide valuable insights into the neural regions influencing diagnostic decisions, aiding clinicians and researchers in understanding the underlying mechanisms driving classification outcomes. By visualizing the regions of interest identified by the model, clinicians can gain confidence in its diagnostic recommendations and explore the potential biomarkers associated with Alzheimer’s disease pathology.

### 5.2. Comparison with Previous Studies

This study stands at the forefront of Alzheimer’s disease diagnosis research by integrating deep transfer learning and explainable artificial intelligence (XAI) techniques. A comparative analysis with prior studies in the field highlights the distinctive contributions and advancements of our proposed model (see Table 5).

The proposed method, which employs VGG16, VGG19, DenseNet169, and DenseNet201, as well as two ensembles (VGG16/VGG19 and DenseNet169/DenseNet201), along with EfficientNetB3 and CNNs, achieved a remarkable accuracy of 96% on MRI OASIS scans.

In order to fortify interpretability and transparency in Alzheimer’s diagnosis, our study introduces a novel model that achieves an impressive accuracy of 96%. What distinguishes our model is the integration of cutting-edge XAI techniques, including saliency maps and grad-CAM (gradient-weighted class activation mapping). Unlike some of the existing studies that focus on specific XAI frameworks, such as LIME [15], SHAP [15], GNNExplainer [17], or occlusion sensitivity mapping [18], our research embraces a combination of techniques to provide a comprehensive and visually intuitive understanding of the model’s decision-making process.

When compared to the current methods, we see the following results:

Support vector machines, MLP, and KNN: Achieved 91.4% accuracy using LIME and SHAP on a dementia dataset [15].

CNN: Achieved 95.4% accuracy using HAM and PCR on MRI ADNI scans [16].

Graph neural network (GNN): Yielded an accuracy range of 53.5 ± 4.5% using GNN explainer on Ex-ADNI data [17].

EfficientNetB0: Attained 80% accuracy through occlusion sensitivity mapping on MRI OASIS scans [18].

3D CNN (multiple instances): Performance varies across different approaches and datasets, ranging from 18% to 77% accuracy, using various XAI methods on PET and MRI scans [19].

Various classifiers (KNN, RF, AdaBoost, gradient boosting, Bernoulli NB, and SVM): Achieved 91% accuracy using DT on cognitive and PET images [20].

3D CNN (using various XAI methods): Achieved 76.6% accuracy using 3D ultrametric contour maps, 3D class activation maps, and 3D gradCAM on ADNI scans [21].

3D CNN (with sensitivity analysis occlusion): Achieved 77% accuracy using sensitivity analysis occlusion on MRI and PET scans [22].

## 6. Conclusions and Future Work

### 6.1. Conclusions

This paper presents a pioneering approach to Alzheimer’s disease diagnosis by integrating deep transfer learning and explainable artificial intelligence (XAI) techniques. The extensive experimentation with popular pretrained convolutional neural networks (CNNs) demonstrated the effectiveness of ensembles, specifically Ensemble-1 (VGG16 and VGG19) and Ensemble-2 (DenseNet169 and DenseNet201), in terms of achieving superior diagnostic performance, reaching up to 95% accuracy, precision, recall, and F1 scores. This signifies the potential of leveraging pretrained models and ensemble techniques for enhanced predictive capabilities in Alzheimer’s diagnosis.

In order to address the crucial need for interpretability and transparency in medical diagnostics, we introduced a novel model with an impressive accuracy of 96%. The incorporation of explainable AI methodologies, including saliency maps and grad-CAM, played a pivotal role in enhancing the accuracy of our model. Furthermore, these techniques offered clinicians and researchers insightful visualizations of the neural regions, influencing diagnostic decisions.

### 6.2. Contributions of This Study

In summary, this research significantly contributes to the realm of Alzheimer’s disease detection through the following key contributions:1.The development of a robust deep transfer learning ensemble model for accurate Alzheimer’s disease classification;2.The introduction of a novel diagnostic model that achieved an impressive 96% accuracy;3.The advancement of interpretability and transparency through the integration of explainable AI techniques such as saliency maps and grad-CAM;4.The implementation of cutting-edge XAI methods to provide transparent and intuitive explanations for diagnostic predictions;5.A rigorous performance evaluation using benchmark datasets showcasing the model’s superiority in both accuracy and interpretability when compared to state-of-the-art methods.

### 6.3. Future Work

As we look ahead, there are several avenues for future research and development in this domain. Firstly, further exploration of deep transfer learning ensembles could involve investigating additional pretrained models and their combinations to optimize diagnostic performance. Additionally, refining the explainable AI techniques employed in our model and exploring newer methods might contribute to even more transparent and interpretable diagnostic systems.

Furthermore, the evaluation of the proposed model on diverse and larger datasets would enhance its generalizability and robustness. Collaborations with healthcare institutions for real-world validation and integration into clinical workflows would be crucial to assess the practical utility and impact of the developed model in a clinical setting.

The continuous evolution of AI and medical imaging technologies opens possibilities for incorporating multi-modal data and advanced feature extraction methods. Integrating genetic, demographic, or longitudinal data might potentially improve the accuracy and early detection capabilities of the diagnostic model.

## Figures and Tables

**Figure 1 diagnostics-14-00345-f001:**
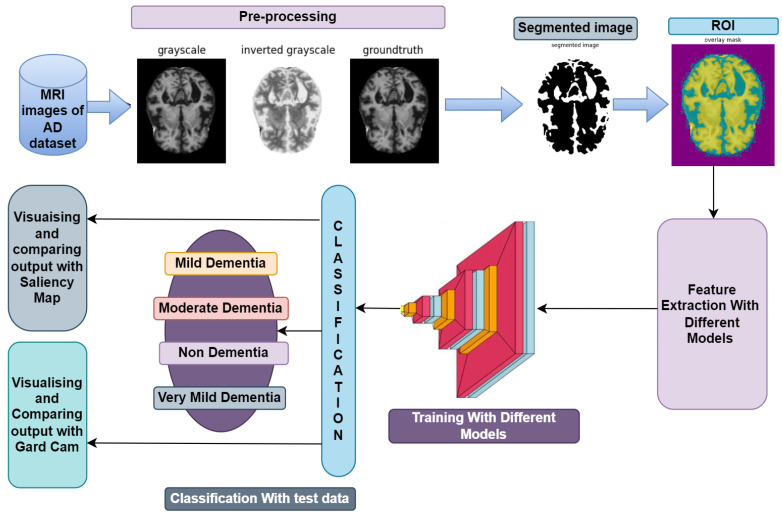
Workflow diagram for diagnosing MRI images for the detection of AD.

**Figure 2 diagnostics-14-00345-f002:**
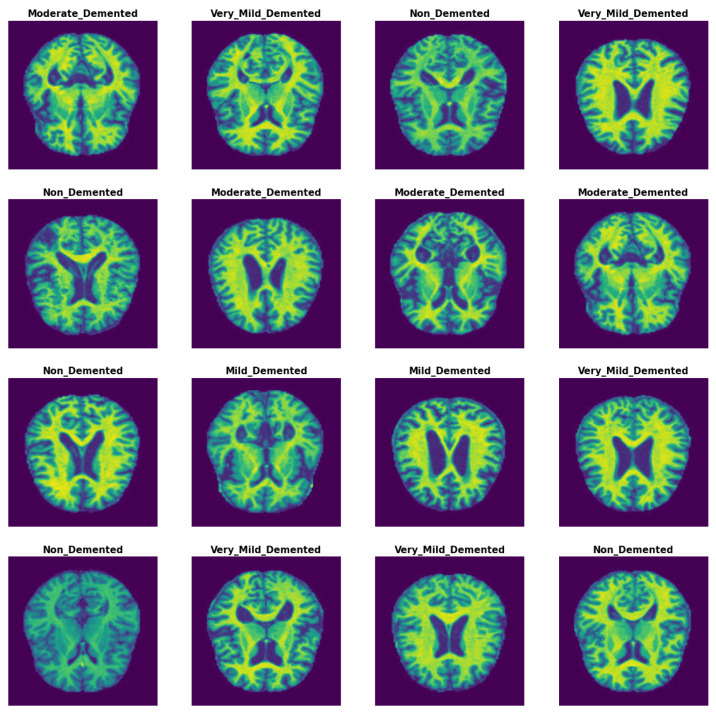
MRI image samples from AD dataset.

**Figure 3 diagnostics-14-00345-f003:**
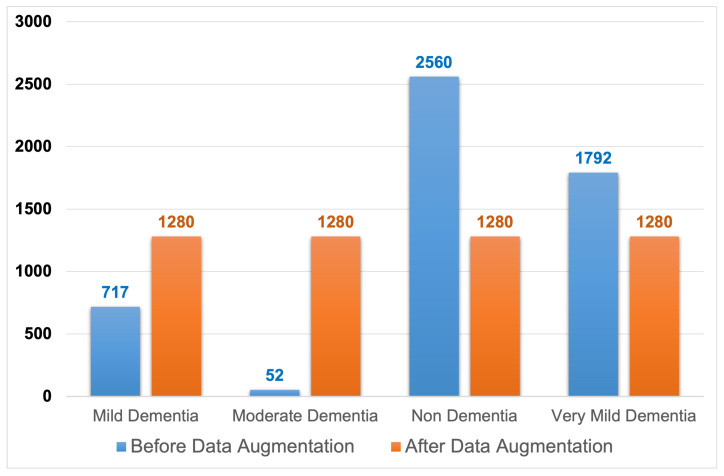
Training dataset before and after data augmentation.

**Figure 4 diagnostics-14-00345-f004:**
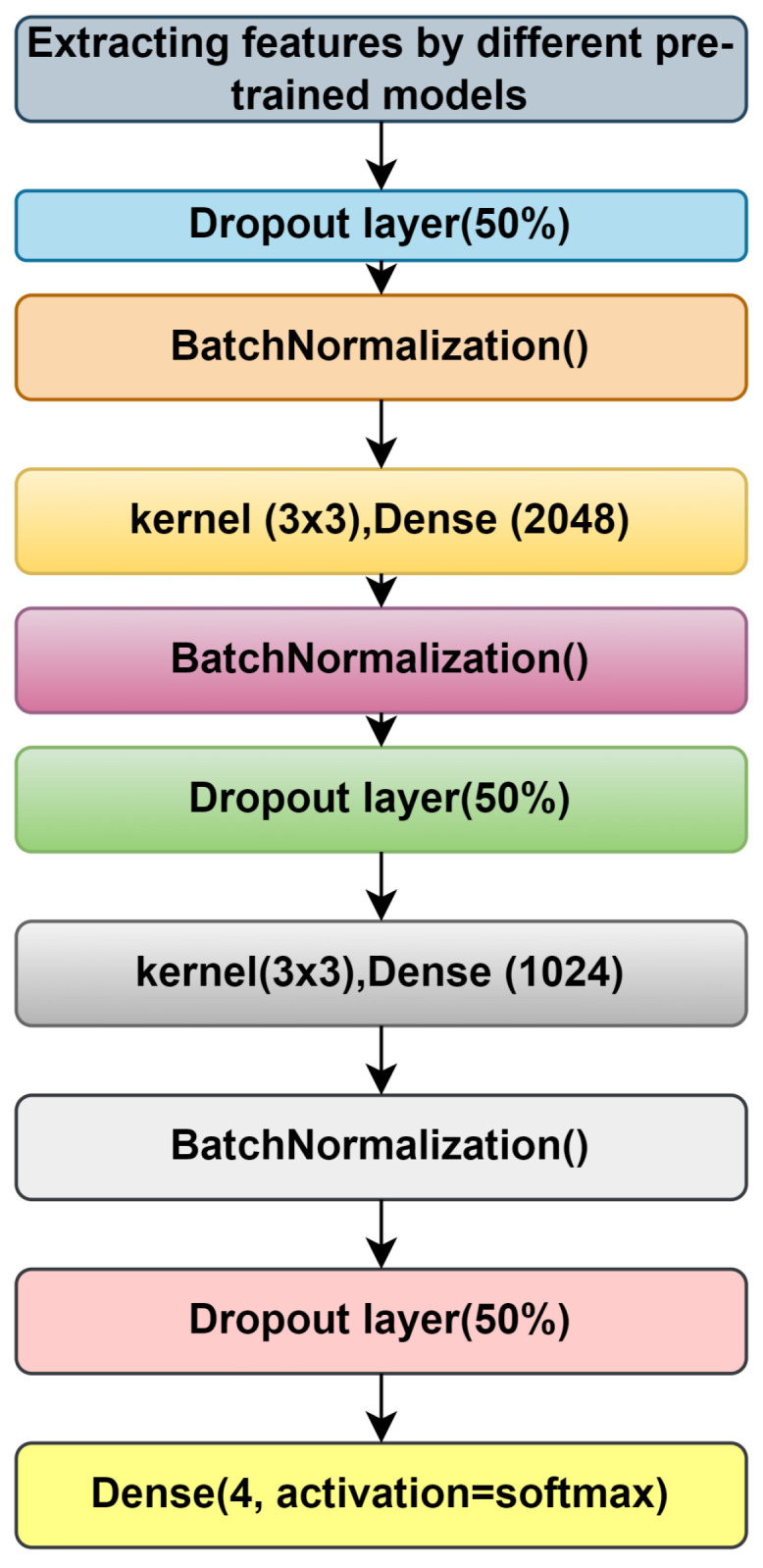
Addition of new layers after feature extraction.

**Figure 5 diagnostics-14-00345-f005:**
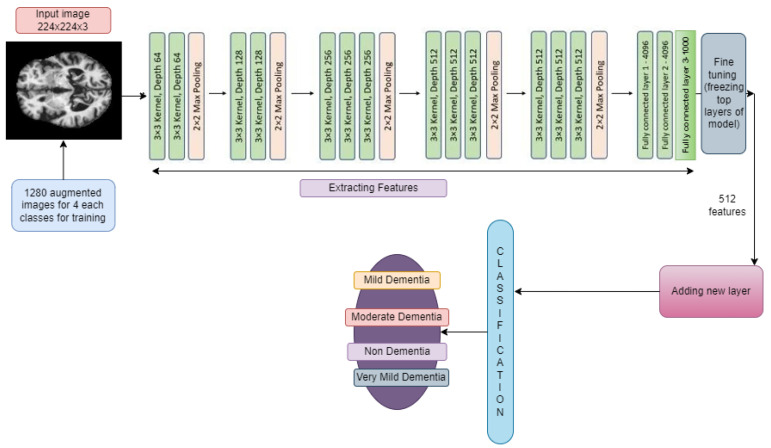
Addition of new layers after feature extraction using VGG16.

**Figure 6 diagnostics-14-00345-f006:**
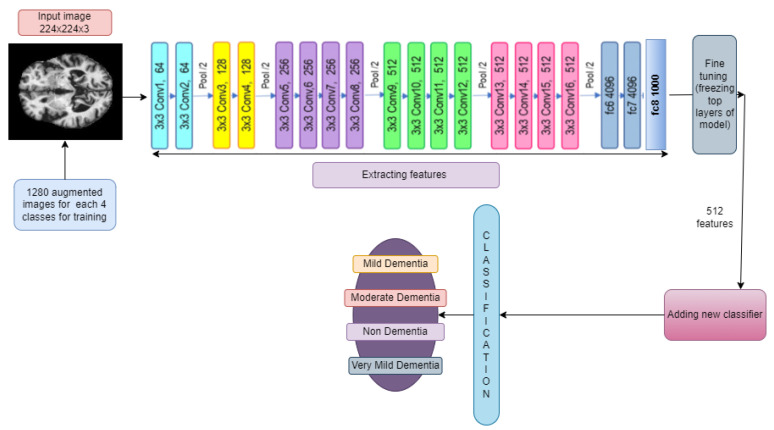
Addition of new layers after feature extraction using VGG19.

**Figure 7 diagnostics-14-00345-f007:**
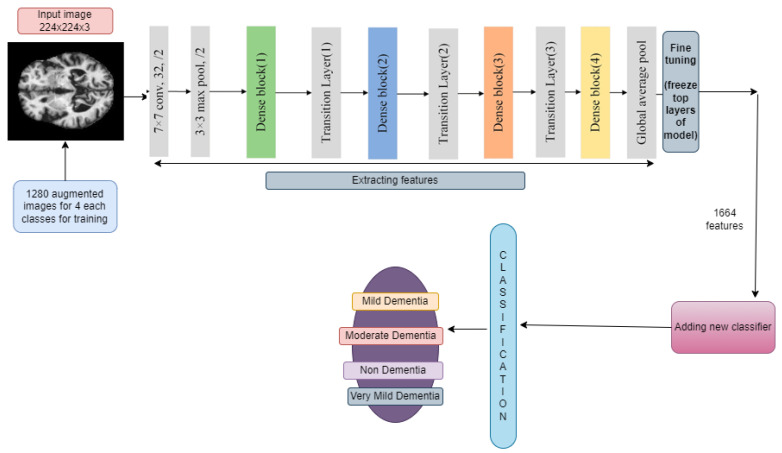
Addition of new layers after feature extraction using DenseNet169.

**Figure 8 diagnostics-14-00345-f008:**
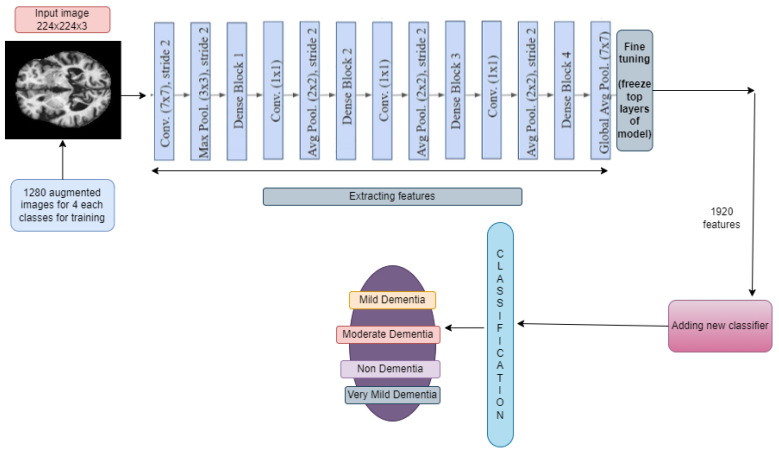
Addition of new layers after feature extraction using DenseNet201.

**Figure 9 diagnostics-14-00345-f009:**
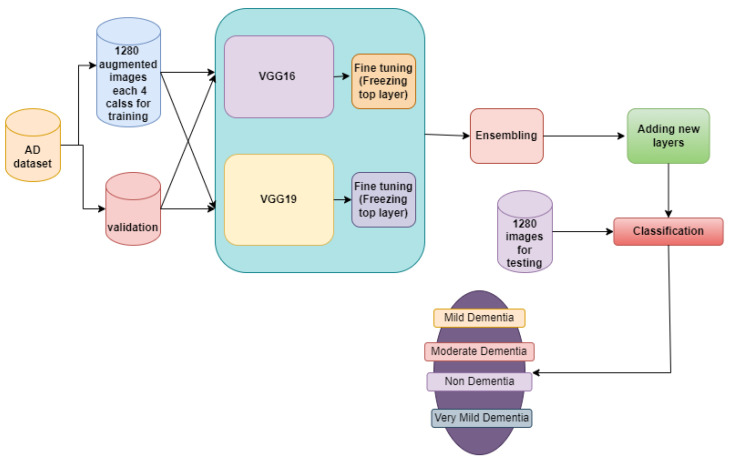
Ensemble-1 model architecture.

**Figure 10 diagnostics-14-00345-f010:**
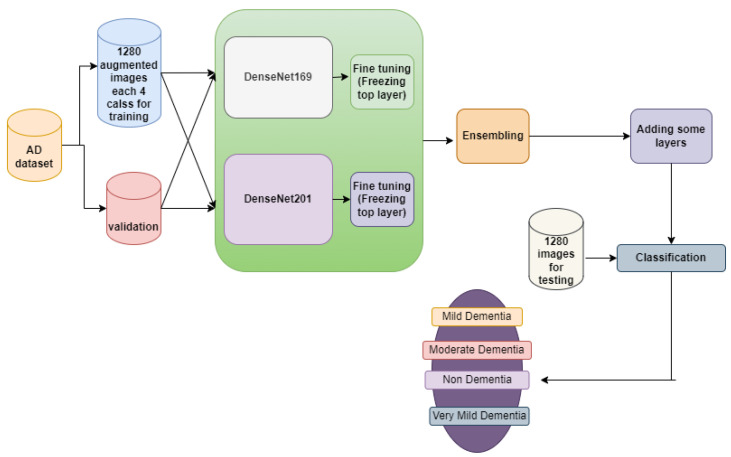
Ensemble-2 model architecture.

**Figure 11 diagnostics-14-00345-f011:**
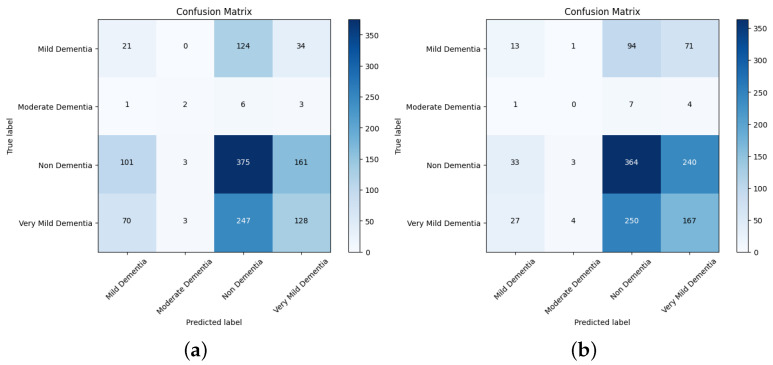
Confusion matrix for VGG16 and VGG19. (**a**) VGG16. (**b**) VGG19.

**Figure 12 diagnostics-14-00345-f012:**
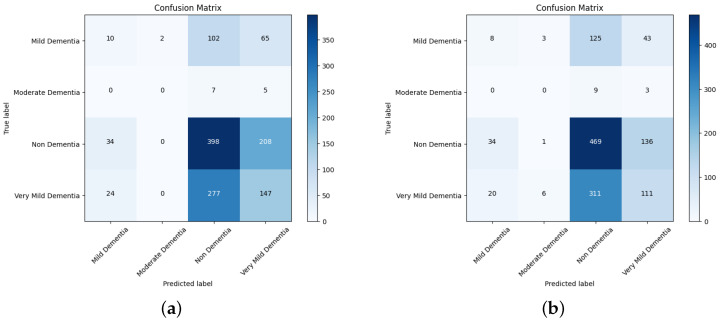
Confusion matrix for DenseNet169 and Densenet201. (**a**) DenseNet169. (**b**) DenseNet201.

**Figure 13 diagnostics-14-00345-f013:**
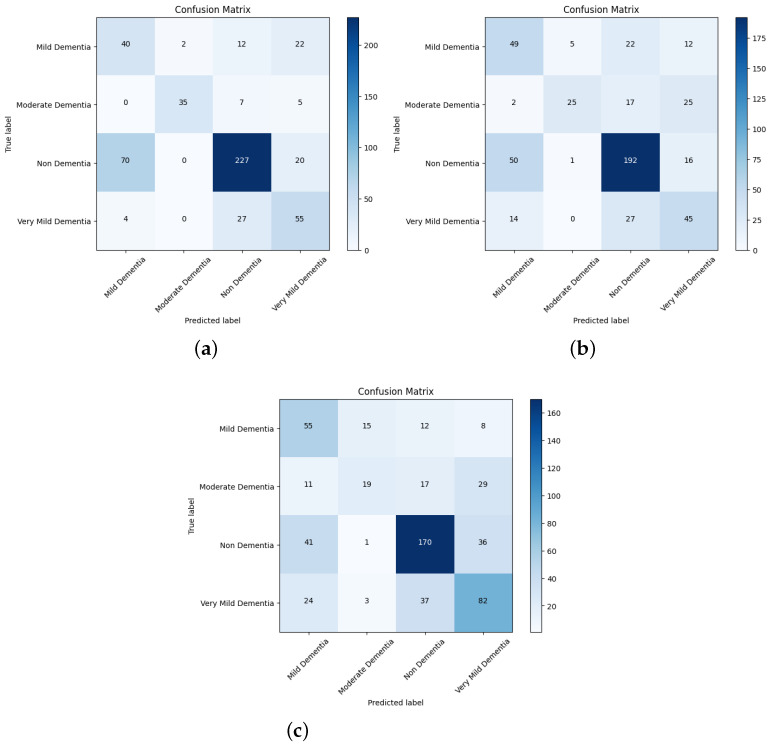
Confusion matrix for Ensemble-1, Ensemble-2, and the proposed model. (**a**) Ensemble-1. (**b**) Ensemble-2. (**c**) Proposed model.

**Figure 14 diagnostics-14-00345-f014:**
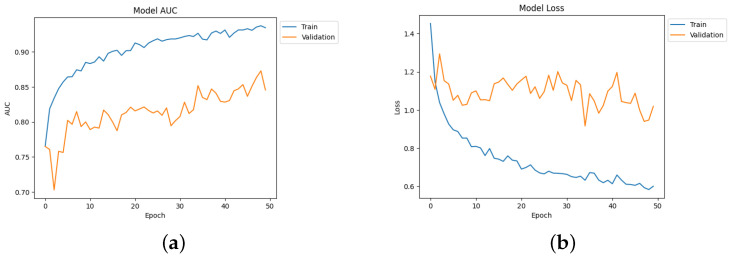
Accuracy and loss curve for VGG16. (**a**) Accuracy. (**b**) Loss.

**Figure 15 diagnostics-14-00345-f015:**
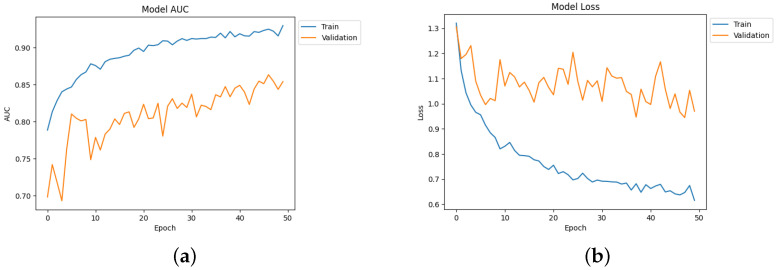
Accuracy and loss curve for VGG19. (**a**) Accuracy. (**b**) Loss.

**Figure 16 diagnostics-14-00345-f016:**
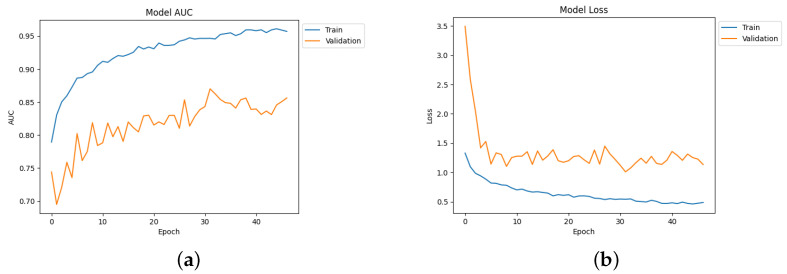
Accuracy and loss curve for DenseNet169. (**a**) Accuracy. (**b**) Loss.

**Figure 17 diagnostics-14-00345-f017:**
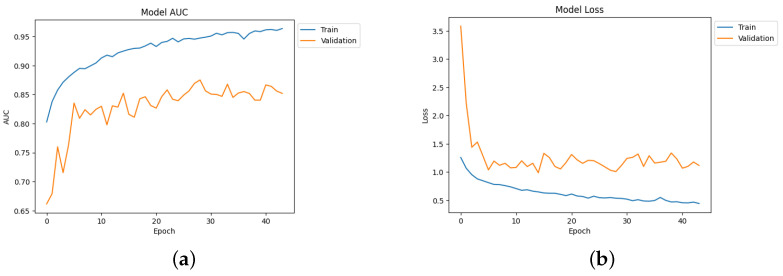
Accuracy and loss curve for DenseNet201. (**a**) Accuracy. (**b**) Loss.

**Figure 18 diagnostics-14-00345-f018:**
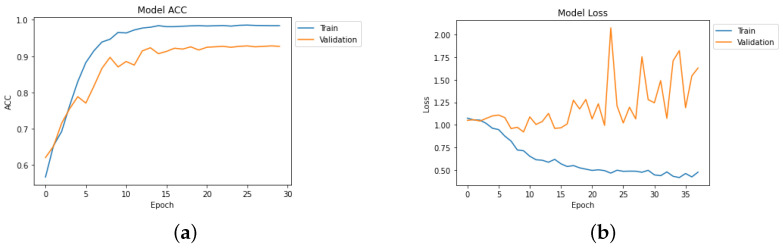
Accuracy and loss curve for Ensemble-1. (**a**) Accuracy. (**b**) Loss.

**Figure 19 diagnostics-14-00345-f019:**
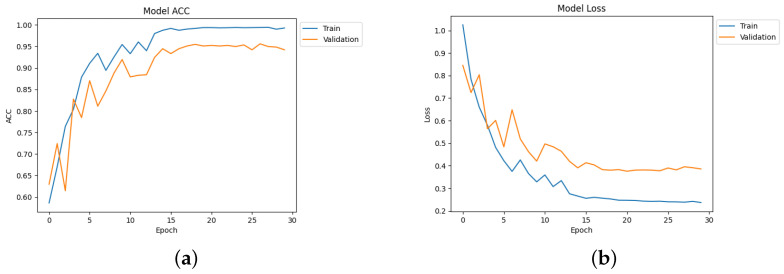
Accuracy and loss curve for Ensemble-2. (**a**) Accuracy. (**b**) Loss.

**Figure 20 diagnostics-14-00345-f020:**
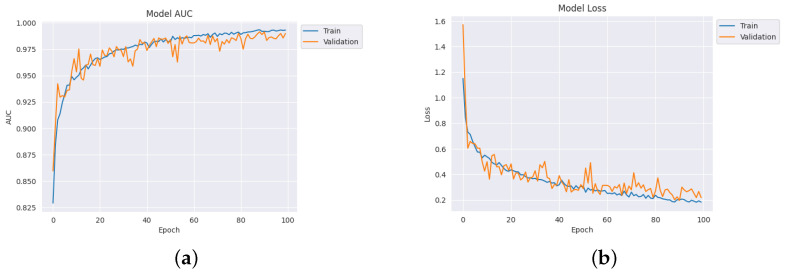
Accuracy and loss curve for the proposed model. (**a**) Accuracy. (**b**) Loss.

**Figure 21 diagnostics-14-00345-f021:**
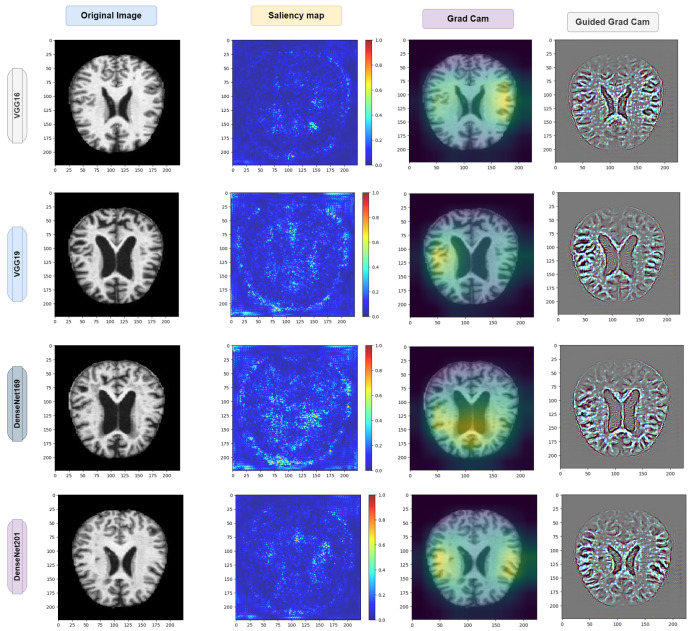
Visual depiction of the saliency map and grad-CAM results for the pretrained model.

**Figure 22 diagnostics-14-00345-f022:**
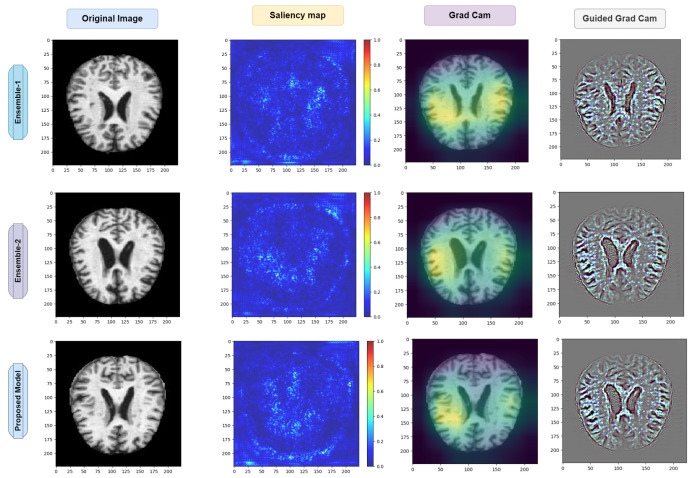
Visual depiction of the saliency map and grad-CAM results for the ensemble and proposed model.

**Table 1 diagnostics-14-00345-t001:** Dataset statistics.

Class Name	Number of Images
Mild Dementia	896
Moderate Dementia	64
Non-Dementia	3200
Very Mild Dementia	2240

**Table 2 diagnostics-14-00345-t002:** Training dataset before and after data augmentation.

Class Name	Before Data Augmentation	After Data Augmentation
Mild Dementia	717	1280
Moderate Dementia	52	1280
Non-Dementia	2560	1280
Very Mild Dementia	1792	1280

**Table 3 diagnostics-14-00345-t003:** Description of the proposed model architecture.

Model Content	Details
Input Image Size	224 × 224 × 3, with 5120 training images and 1280 images in each class
Feature extraction	Using EfficientNet with 1280 features
First Convolution Layer	32 filters; size = 3 × 3; ReLu; Padding = ‘Same’
First Max Pooling Layer	Pooling Size: 2 × 2
Second Convolution Layer	64 filters; size = 3 × 3; ReLu; Padding = ‘Same’
Second Max Pooling Layer	Pooling size: 2 × 2
Third Convolution Layer	128 filters; size = 3 × 3; ReLu; Padding = ‘Same’
Third Max Pooling Layer	Pooling size: 2 × 2
Fourth Convolution Layer	256 filters; size = 3 × 3; ReLu; Padding = ‘Same’
Fourth Max Pooling Layer	Pooling Size: 2 × 2
Fifth Convolution Layer	512 filters; size = 3 × 3; ReLu; Padding = ‘Same’
Fifth Max Pooling Layer	Pooling Size: 2 × 2
Fully Connected Layer	4096 nodes; ReLU
Dropout Layer	50% Neurons dropped randomly
Dense_1 Layer	8320 nodes; ReLu
Dense_2 Layer	516 nodes; ReLu
Output Layer	Four nodes; Softmax activation
Optimization Function	Adam optimization
Learning Rate	0.001
Loss Function	Categorical cross entropy

**Table 4 diagnostics-14-00345-t004:** Model performance.

Model_Name	Accuracy	Precision	Recall	F1-Score
VGG16	90%	89%	84%	86%
VGG19	89%	86%	82%	84%
DenseNet169	87%	85%	84%	83%
DenseNet201	88%	86%	81%	85%
Ensemble-1	92%	90%	88%	87%
Ensemble-2	95%	91%	90%	89%
Proposed model	96%	89 %	93%	91%

**Table 5 diagnostics-14-00345-t005:** Comparison of the proposed method with the state-of-the-art.

Paper	Classifier	Best Score (Accuracy)	XAI Method	Dataset
[15]	Support Vector Machines, KNN, MLP	91.4%	LIME, SHAP	Dementia dataset
[16]	CNN	95.4%	HAM, PCR	MRI scans ADNI
[17]	Graph Neural Network (GNN)	53.5 ± 4.5%	GNN Explainer	ADNI
[18]	EfficientNetB0	80%	Occlusion Sensitivity Mapping	MRI scans OASIS
[19]	3D CNN	-	Saliency Map, LRP	18F-FDG PET
[20]	KNN, RF, AdaBoost, Gradient Boosting Bernouli NB, SVM	91%	DT	Cognitive and and PET images
[21]	3D CNN	76.6%	3D Ultrametric Contour Map, 3D Class Activation Map, 3D GradCAM	ADNI
[22]	3D CNN	77%	Sensitivity Analysis Occlusion	MRI, PET
Proposed Method	VGG16, VGG19, DenseNet169, DenseNet201, Ensemble 1 (VGG16, VGG19) Ensemble 2 (DenseNet169, DenseNet20), Proposed model (EfficientNetB3 & CNN)	96%	Saliency maps, Grad-CAM (Gradient-weighted Class Activation Mapping)	MRI scans OASIS

## Data Availability

The data used to support the findings of this study are available upon reasonable request to the corresponding author.

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
