# Peer review of "An Explainable AI Paradigm for Alzheimer’s Diagnosis Using Deep Transfer Learning"

_diagnostics, 2024, doi:10.3390/diagnostics14030345_

Round 1
Reviewer 1 Report
Comments and Suggestions for Authors
In this study recommended for Alzheimer's disease, many methods were used and high success was achieved. But there are a few problems.
Superficial problems:
Figures 10,11,12,13 and 14 should be more understandable, and the names of the values and attributes in the images should be visible.
Figures 3 and 4 must be original images, and Figure 3 must be made with a more professional tool.
In terms of quality:
The results of this study should be shown in a comparative table with the results of similar studies in the literature.
The difference between this study and those in the literature should be clearly stated.
The contribution of this study to the literature should be given, and its difference from other studies (success, duration, cost, etc.) should be given.
Discussion section should be improved
The success of the study should be mentioned in the results section.
Comparisons should be made with the same datasets or different datasets
Comparisons should be made with the same or different models.
In Figure 10, DendeNet should be written as DenseNet.
Comments on the Quality of English LanguageIn this study recommended for Alzheimer's disease, many methods were used and high success was achieved. But there are a few problems.
Superficial problems:
Figures 10,11,12,13 and 14 should be more understandable, and the names of the values and attributes in the images should be visible.
Figures 3 and 4 must be original images, and Figure 3 must be made with a more professional tool.
In terms of quality:
The results of this study should be shown in a comparative table with the results of similar studies in the literature.
The difference between this study and those in the literature should be clearly stated.
The contribution of this study to the literature should be given, and its difference from other studies (success, duration, cost, etc.) should be given.
Discussion section should be improved
The success of the study should be mentioned in the results section.
Comparisons should be made with the same datasets or different datasets
Comparisons should be made with the same or different models.
In Figure 10, DendeNet should be written as DenseNet.
Reviewer 2 Report
Comments and Suggestions for Authors
Currently, the whole world is facing the problem of aging population. Unfortunately, this leads to an increase in the number of people suffering from neurodegenerative diseases. Despite the prevalence of Alzheimer's disease, its diagnosis is often overlooked or misdiagnosed. Therefore, the presented problem is relevant and has promising applications. Another challenge is early diagnosis. Although there is no cure for Alzheimer's disease, early detection of signs of dementia or depression can help an aging person maintain cognitive function.
And the third issue raised in this manuscript is the use of new technologies - neural networks. The authors claim that the proposed model simplifies the work of a physician, it is accurate and has many functions.
In my opinion, the article is well written. In the introduction, the authors managed to logically derive the aim and objectives of the study from the current literature.
The proposed methods do not raise any questions. All justifications are given, the chains are logically built. There are no principal remarks. The work is well organized.
There are no principle comments. The article is well designed.
Figure 3 raises some doubts. Perhaps it is worth to change its configuration or remove it. In principle, it does not carry much information, as the data are described in the text and shown in Table 2.
Discussion. Perhaps the section "Comparison with previous studies" should characterize previous studies. This section is now more about the authors' own model, although it is certainly a good one.
The conclusions drawn by the authors are consistent with the results obtained. A large block of information has been analyzed and the conclusions are certainly credible.
Overall, I would like to congratulate the authors for good results and an excellent article. I hope that their methodology will improve the lives of both patients and their physicians.
Reviewer 3 Report
Comments and Suggestions for Authors
The authors of the manuscript describe an original clinical study aimed at solving the urgent task of finding new diagnostic methods based on neuroimaging and data analysis by artificial intelligence.
Despite the shortcomings in the style of writing the manuscript, the data obtained are of high importance for practical healthcare. The results shown by the authors can make a significant contribution to the development of diagnostic algorithms for classifying Alzheimer's disease and helping neurologists. Therefore, the conducted research will be relevant for publication in the journal Diagnostics.
However, during the review of the manuscript, several points were found that can be improved:
1. It is necessary to add information about the protocol of the ethics committee for this study.
2. In the "Introduction" section, the authors use overly bold (even advertising) theses describing the developed software solution. I recommend changing the spelling to a restrained scientific format.;
3. In the "Introduction" section, much attention is paid to the description of the developed solution, instead of analyzing the relevance and considering the existing situation in this area (it is very superficially executed);
4. I highly recommend that the authors combine section 1 and section 2 into a single section "Introduction", removing the description of the structure of the manuscript from them (lines 67-72). It is also recommended to shorten the text in lines 56-66
5. It is strange that in the manuscript the section "results" and the section "Discussion" are combined into one called "Experimental Results". I recommend highlighting a separate section "5. Discussion".
6. In the section "5.1. Conclusion" the authors again use very lofty statements about their developed software solution. I recommend changing the spelling to a more restrained style.
